# Association of poor housing conditions with COVID-19 incidence and mortality across US counties

**Khansa Ahmad, Sebhat Erqou, Nishant Shah, Umair Nazir, Alan R. Morrison, Gaurav Choudhary, Wen-Chih Wu***

The Providence Veterans Affairs Medical Center, Lifespan Hospitals and the Warren Alpert Medical School at Brown University, Providence, Rhode Island, United States of America

* wen-chih.wu@va.gov

**Data Availability Statement:** All relevant data are within the manuscript and its Supporting information files.

## Abstract

### Objective

Poor housing conditions have been linked with worse health outcomes and infectious disease spread. Since the relationship of poor housing conditions with incidence and mortality of COVID-19 is unknown, we investigated the association between poor housing condition and COVID-19 incidence and mortality in US counties.

### Methods

We conducted cross-sectional analysis of county-level data from the US Centers for Disease Control, US Census Bureau and John Hopkins Coronavirus Resource Center for 3135 US counties. The exposure of interest was percentage of households with poor housing conditions (one or greater of: overcrowding, high housing cost, incomplete kitchen facilities, or incomplete plumbing facilities). Outcomes were incidence rate ratios (IRR) and mortality rate ratios (MRR) of COVID-19 across US counties through 4/21/2020. Multilevel generalized linear modeling (with total population of each county as a denominator) was utilized to estimate relative risk of incidence and mortality related to poor housing conditions with adjustment for population density and county characteristics including demographics, income, education, prevalence of medical comorbidities, access to healthcare insurance and emergency rooms, and state-level COVID-19 test density. We report incidence rate ratios (IRRs) and mortality ratios (MRRs) for a 5% increase in prevalence in households with poor housing conditions.

### Results

Across 3135 US counties, the mean percentage of households with poor housing conditions was 14.2% (range 2.7% to 60.2%). On April 21st, the mean (SD) number of cases and deaths of COVID-19 were 255.68 (2877.03) cases and 13.90 (272.22) deaths per county, respectively. In the adjusted models standardized by county population, with each 5% increase in percent households with poor housing conditions, there was a 50% higher risk of

**Funding:** Research reported in this publication was supported by VA HSRD grant I01 HX002422-01A2 (WWC), Research Project Grant NIH NHLBI R01HL139795 (A.R.M.) and an Institutional Development Award (IDeA) from NIH NIGMS P20GM103652 (A.R.M.). This work was also supported by Career Development Award Number 7IK2BX002527 from the United States Department of Veterans Affairs Biomedical Laboratory Research and Development Program (A.R.M.). The views expressed in this article are those of the authors and do not necessarily reflect the position or policy of the Department of Veterans Affairs or the United States government. This work was also supported by a Lifespan CVI Pilot Grant for Faculty (A.R.M.). Dr Erqou is supported by funding from the Department of Veterans Affairs, Veterans Health Administration, VISN 1 Career Development Award. Dr Erqou also received funding from Center for AIDS Research, The Rhode Island Foundation, and Lifespan Cardiovascular Institute.

**Competing interests:** The authors have declared that no competing interests exist.

COVID-19 incidence (IRR 1.50, 95% CI: 1.38–1.62) and a 42% higher risk of COVID-19 mortality (MRR 1.42, 95% CI: 1.25–1.61). Results remained similar using earlier timepoints (3/31/2020 and 4/10/2020).

## Conclusions and relevance

Counties with a higher percentage of households with poor housing had higher incidence of, and mortality associated with, COVID-19. These findings suggest targeted health policies to support individuals living in poor housing conditions should be considered in further efforts to mitigate adverse outcomes associated with COVID-19.

## Introduction

Coronavirus disease-2019 (COVID-19) is a rapidly evolving pandemic caused by the novel enveloped RNA beta-coronavirus, Severe Acute Respiratory Syndrome Coronavirus 2 (SARS--CoV-2) [1]. As of April 26, 2020, SARS-COV-2 has infected over 2 million people worldwide and led to >200k deaths [2]. The infectivity potential of SARS-CoV-2 is a function of its high basic reproductive number ($R_0$) which has been estimated to be from 2.2 to as high as 5.7, as compared to $R_0$ of 1.3 for influenza virus [3–5]. Given the high infectivity and lack of a vaccine so far, one of the important mitigation strategies employed worldwide in order to decrease the rate of spread and 'flatten' the infection curve has been social and physical distancing [3,6]. However, physical distancing can be compromised if the living environment for various reasons, such as poor housing conditions, prevents the individuals from staying isolated. Problems associated with poor housing, reported as severe housing problems by CDC, include overcrowding, high housing cost, lack of kitchen facilities or lack of plumbing facilities [7,8].

Overcrowding has been associated with spread of respiratory illnesses like tuberculosis and influenza which have aerosol and droplet transmissions, both of which are potential modes of transmission for COVID-19 [9–13]. The World Health Organization (WHO) published housing and health guidelines in 2018, identifying poor housing related environmental risk factors including overcrowding, air and water quality and lack of access to adequate plumbing and sanitation, as factors contributing to the burden of infectious diseases including airborne respiratory illnesses [14]. Another feature of poor housing conditions is high cost burden, defined as more than 50% of the household income spent towards housing cost [8]. Households facing high housing cost in proportion to the household income have to compromise in other aspects of living, such as education, health insurance and food, among others [15]. Prior studies have shown an association between high housing costs and delays in seeking healthcare [16,17]. There are several news reports indicating that minimum wage workers have found it difficult to follow the stay-at-home guidelines, due to a lack of economic resources (e.g. public transportation, choice of working environment) and being part of essential work force that is mostly an exception to stay-at-home guidelines [18–20]. Another aspect of poor housing conditions is lack of plumbing or kitchen facilities and the inherent need to utilize communal facilities (bathrooms or kitchens). Lack of appropriate facilities interferes with the ability to practice good hygiene. Based on the above, people with poor housing conditions have a limited ability to practice effective physical distancing and good personal hygiene; hence, potentially at higher risk of worse outcomes related to a respiratory infectious illness such as COVID-19.

Since the relationship between poor housing conditions and COVID-19 outcomes are currently unknown, we undertook a study to test the hypothesis that the percentage of households

with poor housing conditions in US counties are associated with higher incidence and mortality of COVID-19 across 3135 counties in the United States (US). Robust literature had linked poor housing conditions to worse health outcomes [16,21–23]. Conversely, availability of appropriate plumbing facilities and in-home clean water has been linked to a decline in infectious diseases including lower respiratory tract infections [14,24] and highlights the importance of this study.

## Materials and methods

We conducted a cross sectional ecological analysis of the 3141 US counties using publicly available data relating poor housing conditions with COVID-19 outcomes. Counties with missing data for poor housing condition (n = 6) were excluded yielding a sample size of 3135 US counties. These counties were Prince of Wales-Outer Ketchikan, Skagway-Hoonah-Angoon, Wrangell-Petersburg and Kusilvak from Alaska, Bedford City from Virginia and Oglala Lakota from South Dakota. Counties in US territories (American Samoa, Guam, Northern Mariana Island, Puerto Rico and US Virgin Islands) were not included in our analysis. Since there is no individual identifying information, the data were in aggregate by county, and publicly available from the Centers for Disease Control (CDC), US census Bureau and John Hopkins Coronavirus Resource Center, the protocol received exemption from the Providence Veterans Affairs Medical Center Institutional Review Board [2,25,26].

### Exposure

The exposure of interest was the percentage of households in a county with poor housing conditions which has been published as percentage of households with severe housing problems (2010–2014) by the CDC [27]. These households were identified as having any of the four problems: 1) overcrowding, 2) high housing cost burden, 3) incomplete kitchen facilities, and 4) incomplete plumbing facilities. In further breakdown, overcrowding is defined as more than one person per room, high housing cost as more than 50% of the monthly household income allocated towards housing cost (including utilities), incomplete kitchen facilities as lacking a sink with running water, stove or range, or a refrigerator, and incomplete plumbing facilities as lacking of hot and cold piped water, a flush toilet, or a bathtub/shower [8]. To facilitate interpretation, we studied the poor housing conditions as a continuous variable by units of 5% increase in households with poor housing conditions, which is roughly equivalent to the standard deviation (SD) of the variable. Additionally, we also categorized the counties according to approximate quartiles based on percentage of households with poor housing conditions (rounded values were used as cut off).

### Outcome

The outcomes of interest were incidence relative risk and mortality relative risk of COVID-19 related to poor housing condition. For this purpose, we obtained the COVID-19 cases and death data from the John Hopkins Coronavirus Resource Center, where it has been published for "public health, educational, and academic research purposes." [2] We obtained the cumulative data for three dates of 10-day intervals: March 31st, April 10th and April 21st, which allowed us to test the robustness of our findings across three temporal cross sections of the reported data. April 21st data were utilized in the main analysis. These numbers of cases and deaths per county were subsequently incorporated into an equation with the total population of each county as denominator for standardization to incidence (cases/100,000) and mortality (deaths/100,000) in the regression modelling [26], to calculate the incident rate ratio and mortality rate ratio related to poor housing conditions.

## Covariates

We obtained data on county level variables related to COVID-19 spread and outcomes based on literature. The data regarding total population (2010) and population density (population per square miles of land area of a county) were obtained to account for the exposure pool [27]. In order to account for socioeconomic disparity, data on median household income of a county (2016) and percentage of residents without a high school diploma (2013–2017) were included [27].

County demographic data was collected because male sex, older age and percent of racial minority have been linked to a higher risk of, and mortality in, COVID-19 [28–31]. This included percentage of male residents (2010), median age (2010), and percentage of white, black, Hispanic or Latino, Asian, Native Hawaiian or Pacific Islander; and American Indian or Alaska Native residents (2013–2017) [27].

From the very initial reports, COVID-19 was seen to drastically effect individuals with a heavier burden of comorbidities [28,29]. We therefore collected data for percentage of residents diagnosed with diabetes and those with obesity (2015) [27]. Furthermore, Medicare hospitalization data for: hypertension, ischemic stroke, myocardial ischemia, heart failure and dysrhythmia (2014–2016) were obtained as a surrogate for cardiovascular disease burden in the county [27].

The morbidity and mortality associated with COVID-19 has been linked to respiratory failure [32]. We therefore collected variables that affect respiratory health: annual concentration of particulate matter 2.5 μ (pm2.5) (2014) and percentage of residents who are current smokers (2017) [27].

We accounted for the percentage of adults without health insurance under 65 years (2016) and number of hospitals with emergency rooms (ER) (2016) in each county, as surrogates of access to care. The number of hospitals may also influence the number of cases detected in the county. Furthermore, we obtained the total number of tests conducted per state (cumulative up to April 21st) and calculated the ratio of tests to the total population of the state (test density) [33].

## Statistical analysis

County covariates were described as mean ± SD and range for continuous variables and as number (%) for categorical variables. Linear regression was used to assess linear trend of the county covariates across the four quartiles.

We utilized multilevel generalized linear models with a negative binomial distribution family and a log link function (with county population as a denominator) to determine the association between county-level prevalence of poor housing conditions and the county-level incidence and mortality of COVID-19 standardized to the population of each county. To account for clustering effect due to policy, social and behavioral similarities across counties within the same state, we applied a random intercept for the state. The covariance matrix was specified as unstructured. We adjusted for different categories of variables in the regression model in a stepwise fashion: 1) population density and test density, 2) demographics (% male, median age, % white), 3) socioeconomic status (median household income, % residents with lack of high school education), 4) respiratory exposure (annual ambient PM2.5, % current smokers), 5) prevalence of comorbidities (% diagnosed with diabetes, % diagnosed with obesity), 6) Medicare hospitalization rates (hypertension, ischemic stroke, myocardial ischemia, heart failure, dysrhythmia), and 7) Access to healthcare (% adults without health insurance under 65 years and number of hospitals with ER). The fully adjusted model included all the aforementioned variables. We reported incidence rate ratios (IRR) and mortality rate ratios

(MRR) and 95% confidence intervals (CIs), respectively, which are interpretable as the relative increase in incidence and mortality rates for COVID19, for each 5% increase in households with poor housing conditions.

We conducted several sensitivity analyses to assess the robustness of our findings. 1) In lieu of percent white population in a county, we used the percent breakdown of the minority population: black, Hispanic or Latino, Asian or Native Hawaiian or Pacific Islander and American Indian or Alaska Native residents per county in the fully adjusted model. 2) Quartile Analysis —we also studied quartiles of percent households with poor housing conditions. We tested the quartiles in the fully adjusted model as an ordinal variable (i.e. linear effect across quartiles) and as categorical (dummy) variables (using 1st quartile as referent). 3) Temporal Analysis— Since the main analysis is using the most recent data (April 21st, 2020), we repeated our analyses in two earlier time points for COVID-19 incidence and mortality as outcomes, on March 31st and April 10th, to account for and understand temporal changes, if any, in this association.

A two-sided p value of <0.05 was considered statistically significant. All analyses were conducted in Stata SE statistical software (Stata Corp, Texas, v. 15.0).

## Results

Across 3135 US counties, the mean (range) percentage of households with poor housing conditions was 14.2% (2.7% to 60.2%). The modified quartiles of percentage of households with poor housing conditions were 2.7% to 11% (Quartile1), 11.1% to 14.0% (Quartile2), 14.1% to 17.0% (Quartile3) and 17.1% to 60.2% (Quartile4). Till April 21st, there were a total of 144190 confirmed COVID-19 cases assigned to these 3135 US counties and 14887 COVID-19 deaths. The mean (SD) number of cases and deaths of COVID-19 were 255.68 (2877.03) cases and 13.90 (272.22) deaths per county, respectively. Table 1 describes characteristics of counties across the US, overall and stratified by quartiles of percentage of households with poor housing conditions.

The mean number of COVID-19 cases and deaths per county for all time points, March 31st, April 10th and April 21st increased across increasing quartiles of percentage of households with poor housing conditions (all p's <0.001). Similarly, increasing quartiles of percent households with poor housing conditions were associated with higher county population, population density, percentage of minority residents, percentage of residents without high school diploma, prevalence of diabetes, Medicare hospitalization rates for hypertension, ischemic stroke, heart failure and dysrhythmia, percentage of current smokers, annual PM2.5 levels, percentage of adults <65 years without health insurance and number of hospitals with ER facilities (all p's <0.001). Conversely, a decrease in the median household income and prevalence of obesity was observed across increasing quartiles of percent households with poor housing conditions (both p's <0.001).

Using the COVID-19 data from April 21st, we found that for each 5% increase in poor housing condition per county, there is a 59% increase in the relative risk of COVID-19 incidence (IRR 1.59, 95% confidence interval [CI]: 1.49–1.70) (Table 2). The association was only slightly attenuated after adjustment for an extensive list of county covariates (Table 2). The IRR for the fully adjusted model (Model VIII) was 1.50 (95% CI: 1.38–1.62) (Table 2). Secondary analyses categorizing the exposure into quartiles (Fig 1a and 1b) or using data from March 31st and April 10th yielded comparable results (Table 2).

Similarly, using the COVID-19 data from April 21st, we found a 63% increase in the relative risk of COVID-19 mortality for each 5% increase in poor housing condition per county, (MRR 1.63, 95% CI: 1.48–1.79) (Table 3). The association was only mildly attenuated after adjustment (Table 3), and remained highly significant in the fully adjusted model (Model

**Table 1.**

| Variable | Overall Mean ± SD (Range) n = 3135 | Quartiles of Percentage Households with Poor Housing Conditions | | | |
| --- | --- | --- | --- | --- | --- |
| | | Quartile 1(2.7%—11%) Mean ± SD (Range)n = 679 | Quartile 2(11.1%—14%) Mean ± SD(Range)n = 912 | Quartile 3(14.1%—17%) Mean ± SD(Range)n = 830 | Quartile 4(17.1—60.2%) Mean ± SD(Range) n = 714 |
| Total Population (2010) | 98418.71 ± 313273.6 (82–9818605) | 20417.61 ±35194.54 (478–365169) | 51102.02 ± 82563.37 (286–1223348) | 93130.64 ± 158212.9 (662–1503085) | 239180.8 ±603202.7 (82–9818605) |
| Population Density, n/100,000 | 345.18 ± 3792.37 (0.04–168127.7) | 43.16 ± 110.13 (0.16–2014.21) | 183.96 ± 1372.22 (0.17–37333.46) | 581.41 ± 6459.65 (0.09 ± 168127.7) | 560.31 ± 3459.51 (0.04–79500.17) |
| Number of Tests (State level) | 91539.76 ± 101898.8 (7618–633861) | NA | NA | NA | NA |
| Test Density (number of tests/ total population) (State Level) | ± 0.01 (0.01–0.04) | NA | NA | NA | NA |
| Male (2010), % | 49.97 ± 2.22(43.2–72.1) | 50.37 ± 2.30(44.8–68.3) | 49.89 ± 1.90(44.5–64.7) | 49.90 ± 2.27(43.2–72.1) | 49.78 ± 2.41(45.3–66.9) |
| Median Age (2010), years | 40.34 ± 5.04(22.4–62.7) | 42.53 ± 4.27(23.5–55.8) | 41.13 ± 4.26(26.3–62.7) | 39.78 ± 4.53(24–54) | 37.91 ± 5.97(22.4–55.9) |
| White (2013–2017), % | 76.78 ± 20.04(0.6–100) | 87.3 ± 13.27(5.8–100) | 83.01 ±14.07(11.3–99.8) | 75.73 ±17.95(12.8–99.1) | 60.03 ±23.10 (0.6–99.3) |
| Black (2013–2017), % | 8.87 ± 14.45(0–86.9) | 2.72 ± 6.60(0–60.2) | 6.26 ±9.94(0–71.9) | 10.52 ± 14.34 (0–86.3) | 16.12 ± 20.27(0–86.9) |
| Hispanic/Latino (2013–2017), % | 9.08 ± 13.72(0–99.1) | 6.60 ± 11.22(0–93.8) | 6.57 ± 10.15(0–88.4) | 8.78 ± 12.75(0–86.5) | 14.97 ± 18.39(0–99.1) |
| Asian Population or Native Hawaiian or Other Pacific Islander Population (2013–2017), % | 1.36 ± 3.10(0–62.7) | 0.56 ± 0.83(0–6) | 0.85 ± 1.39(0–17.3) | 1.28 ± 2.08(0–27.4) | 2.85 ± 5.55(0–62.7) |
| American Indian/Alaska Native (2013–2017), % | 1.67 ± 7.11(0–82.2) | ± 3.07 (0–54.6) | 1.12 ± 3.53 (0–39.6) | 1.32 ± 4.64 (0–55.7) | 3.56 ± 13.36 (0–90.3) |
| Diagnosed Diabetes, Age-Adjusted Percentage, 20+, (2015), % | 9.83 ± 2.24(3.6 ± 18.3) | 9.11 ± 1.96 (4.6–16.6) | 10.05 ± 2.01 (5.1–17.4) | 10.19 ±2.16 (4.1–16.6) | 9.84 ± 2.68 (3.6–18.3) |
| Obesity, Age-Adjusted Percentage, 20+. (2015), % | 31.96 ± 4.72 (13.5 ± 49.7) | 32.40 ± 3.77 (18.3–46.1) | 32.72 ± 3.91 (16.7–43.8) | 32.24 ±4.55 (14.7–49.7) | 30.25 ± 6.07 (13.5–46.9) |
| Acute Myocardial Infarction Hospitalization (2014–2016) Rate per 1,000 Medicare Beneficiaries | 11.82 ± 3.81(3.6–36.1) | 11.05 ± 3.58 (3.9–31.5) | 12.33 ± 3.92 (3.6–36.1) | 12.29 ±3.85 (4.8–29.2) | 11.35 ± 3.64 (5–26.9) |
| Dysrhythmia Hospitalization (2014–2016) Rate per 1,000 Medicare Beneficiaries | 58.52 ± 16.91 (16.6 ± 136.9) | 51.59 ± 17.13 (16.6–114.6) | 60.46 ± 16.72 (16.9–119.5) | 62.64 ±16.28 (17.3–136.9) | 57.87 ±15.55 (17.3–116.1) |
| Heart Failure Hospitalization (2014–2016) Rate per 1,000 Medicare Beneficiaries | 50.89 ± 19.60 (9.3–148.3) | 43.30 ± 18.82 (9.3–131.2) | 52.70 ± 18.91 (9.9–138.2) | 55.34 ± 19.47 (11.1–138.3) | 50.65 ± 18.94 (10.2–139.8) |
| Hypertension Hospitalization (2014–2016) Rate per 1,000 Medicare Beneficiaries | 132.10 ± 40.01 (32.1–283.2) | 114.20 ± 38.47 (42.4–254.1) | 135.23 ± 37.93 (38.6–276.2) | 141.29 ± 39.33 (38.5–283.2) | 134.47 ± 39.64 (32.1–249.5) |
| Ischemic Stroke Hospitalization (2014–2016) Rate per 1,000 Medicare Beneficiaries | 12.68 ± 3.39 (3.6–35.7) | 11.61 ± 3.51 (4.4–35.7) | 13.13 ± 3.32 (4–33.8) | 13.33 ± 3.30 (3.6–27.8) | 12.36 ± 3.17 (3.7–22.4) |
| Median Household Income (2016), In thousands of $ | 49.52 ± 12.89 (22–134.6) | 51.56 ± 11.61 (24.5–114.7) | 49.39 ± 12.23 (24.9–134.6) | 48.67 ± 12.81 (23.1–115.5) | 48.75 ± 14.65 (22–110.8) |
| Residents without High School Diploma, Ages 25+, (2013–2017), % | 13.81 ± 6.48 (1.1–58.7) | 11.65 ± 5.89 (1.1–47.5) | 13.64 ± 5.95 (1.8–58.7) | 14.32 ± 6.19 (1.9–41.8) | 15.50 ± 7.36 (1.3–51.2) |
| Adults who are current smokers (2017), % | 17.45 ± 3.58 (6–41) | 16.49 ± 3.11 (7–31) | 17.69 ± 3.13 (8–27) | 17.93 ± 3.42 (6–30) | 17.51 ± 4.46 (8–41) |

(*Continued*)

**Table 1.** (Continued)

| Variable | Overall Mean ± SD (Range) n = 3135 | Quartiles of Percentage Households with Poor Housing Conditions | | | |
|---|---|---|---|---|---|
| | | Quartile 1(2.7%—11%) Mean ± SD (Range)n = 679 | Quartile 2(11.1%—14%) Mean ± SD(Range)n = 912 | Quartile 3(14.1%—17%) Mean ± SD(Range)n = 830 | Quartile 4(17.1—60.2%) Mean ± SD(Range) n = 714 |
| Particulate Matter <2.5 μm (2014) Annual Average Ambient Concentrations, ug/m3 | 9.02 ± 1.97 (3–19.7) | 8.39–2.12 (3.4–12.6) | 9.32 ± 1.84 (3–13.3) | 9.21 ± 1.77 (3.4–14.3) | 9.03 ± 2.05 (3.7–19.7) |
| Adults without Health Insurance, Under Age 65, (2016), % | 11.12 ± 4.92 (2.1–33.5) | 9.94 ± 4.83 (2.9–31.1) | 10.68 ± 4.75 (3.0–30.9) | 11.37 ± 4.65 (2.1–33.5) | 12.51 ± 5.18 (2.6–31.0) |
| Hospitals with Emergency Department (2016) | 1.15 ± 1.89 (0–46) | 0.68 ± 0.70 (0–4) | 0.92 ± 0.99 (0–15) | 1.13 ± 1.27 (0–10) | 1.91–3.36 (0–46) |
| **COVID 19 Cases per County (Up to 3/31/2020)** | 58.04 ± 836.02 (0–43119) | 3.10 ± 11.48 (0–141) | 8.82 ± 23.09 (0–325) | 27.17 ± 98.98 (0–1591) | 207.31 ± 1733.53 (0–43119) |
| **COVID-19 Cases per County (Up to 4/10/2020)** | 156.35 ± 1851.53 (0–92384) | 8.95 ± 31.12 (0–431) | 26.58 ± 64.54 (0–825) | 85.00 ± 313.08 (0–4511) | 540.69 ± 3,824.68 (0–92384) |
| **COVID-19 Cases per County (Up to 4/21/2020)** | 255.68 ± 2877.03 (0–144190) | 15.58 ± 51.43 (0–615) | 46.82 ± 117.92 (0–1635) | 155.48 ± 560.40 (0–9621) | 860.10 ± 5,933.84 (0–144190) |
| **COVID-19 Deaths per County (Up to 3/31/2020)** | 0.97 ± 17.36 (0–932) | 0.06 ± 0.33 (0–5) | 0.20 ± 0.70 (0–7) | 0.59 ± 3.12 (0–70) | 3.26 ± 35.98 (0–932) |
| **COVID-19 Deaths per County (Up to 4/10/2020)** | 5.76 ± 107.42 (0–5820) | 0.27 ± 1.25 (0–19) | 0.87 ± 2.63 (0–27) | 2.73 ± 13.53 (0–282) | 20.58 ± 223.13 (0–5820) |
| **COVID-19 Deaths per County (Up to 4/21/2020)** | 13.90 ± 272.22 (0–14887) | 0.62 ± 2.78 (0–39) | 1.93 ± 5.68 (0–74) | 7.44 ± 33.88 (0–506) | 48.91 ± 565.70 (0–14887) |

VIII), with an MRR of 1.42 (95% CI 1.25–1.61) (Table 3). Secondary analyses categorizing the exposure into quartiles (Fig 1a and 1b) or using data from March 31st and April 10th yielded comparable results (Table 3).

## Discussion

To our knowledge, this is the first nationwide study to investigate county level association of COVID-19 incidence and mortality with percentage of households facing poor housing conditions in the US. Our study showed that with each 5% increase in percent households with poor housing conditions, there was a 50% higher risk of COVID-19 incidence and a 42% higher risk of COVID-19 mortality across US counties. Findings remained similar in three different time points and after accounting for county-level population density and state test density, demographics, socioeconomic status, prevalence of comorbidities, respiratory exposure, lack of health insurance and number of ER facilities.

Of the four factors categorized under poor housing conditions, overcrowding and a lack of access to adequate plumbing and sanitation offered the most direct explanation for the higher incidence and mortality of COVID-19. The 2003 spread of severe acute respiratory syndrome (SARS) epidemic in Hong Kong was worst in the Amoy Garden estate which was overcrowded and had significant plumbing and sanitation problems [34]. An initial study from China investigating the initial COVID-19 outbreaks also showed that 79.9% of outbreaks occurred indoors, almost all in apartment settings [35]. Evidence from the Influenza epidemic of 1918 showed not only increased spread but also increased severity of the disease as a result of overcrowding [36]. Overcrowding and inadequate plumbing may lead to repeated exposure and potentially a higher viral inoculum, which had been linked to worse COVID-19 clinical outcomes [37–39]. A more severe COVID-19 disease process may offer a potential explanation of

**Table 2. Relative risk of county Covid-19 incidence as of April 21, 2020 related to percent households with poor housing conditions.**

| Model | Incidence Rate Ratio, IRR (95% Confidence Interval) |
|---|---|
| I- Percentage Households Living with Poor housing conditions | 1.59 (1.49–1.70) |
| II- Model I + Population Density and Test Density | 1.58 (1.48–1.68) |
| III- Model II + Demographics (Male, Age, % White) | 1.31 (1.21–1.42) |
| IV- Model III + Socioeconomic Factors (Median Household Income, % Residents with Lack of High School Education) | 1.48 (1.38–1.60) |
| V- Model IV + Respiratory Exposure Variables (Annual PM25 and Percentage of Residents who reported Smoking) | 1.55 (1.44–1.67) |
| VI- Model V + County Prevalence of Comorbidities (Diabetes and Obesity) | 1.52 (1.41–1.65) |
| VII- Model VI + Hospitalization Rate per 1,000 Medicare Beneficiaries (HF, Dysrhythmia, HTN, Ischemic Stroke, MI) | 1.51 (1.40–1.63) |
| VIII- Model VII + % Adults without insurance + Number of Hospitals with ER/county (Fully Adjusted Model) | 1.50 (1.38–1.62) |
| **Sensitivity Analyses** | |
| IX- Model VIII with Minority Demographics (Male, Age, Black%, Hispanic %, Asian%. Native Americans% in lieu of White%) | 1.43 (1.32–1.54) |
| X- Model VIII with poor housing conditions Quartiles as an Ordinal Variable | 1.25 (1.18–1.32) |
| XI- Model VIII for March 31st, 2020 | 1.65 (1.51–1.81) |
| XII- Model VIII for April 10th, 2020 | 1.24 (1.15–1.34) |

**Note:** Incidence Rate Ratios are from: Multilevel Generalized Linear Negative Binomial Models allowing random intercepts for States. A 5% increase in households with poor housing conditions represents approximately 1-SD change.

the higher mortality. Health education via print or social media targeted at the population at risk to improve awareness of preventative measures and hygiene should be employed to counteract the potential risks in overcrowded facilities [40]. Moreover, to mitigate indoor airborne transmission of COVID-19, investment in engineering controls to improve ventilation, prevent recirculation and using air cleaning filters and disinfecting mechanisms have been proposed to be potential solutions to mitigate indoor transmission, applicable to communal living (apartments), transportation (e.g. bus, train or stations) and work spaces [41]. Moreover, a lack of appropriate plumbing and kitchen facilities in their residence would require the residents to use communal facilities, thereby increasing social contact. It is important to note that SARS-CoV-2 is detectable for up to 72 hours on plastic and stainless steel materials, whereas influenza virus is detectable for only 24–48 hours after [42,43]. This poses another public health risk for transmission to other individuals sharing a crowded space, in the absence of meticulous hygiene. Establishment of hygiene protocols and increased availability of mobile bathrooms and cleaning supplies at communal facilities should also be considered to help mitigate the COVID-19 spread [44].

Another factor of poor housing is high housing cost, which translates into a lack of resources when it comes to seeking healthcare as well as being able to stay at home [15,17,19,45]. Our study showed that counties with the highest percentage of residents with poor housing conditions also had the lowest median household income and highest percentage of residents without a high school diploma. However, an important feature of our study is that although linked to economic status, the associations between poor housing conditions and COVID-19 incidence, and mortality, were independent of the racial composition, median

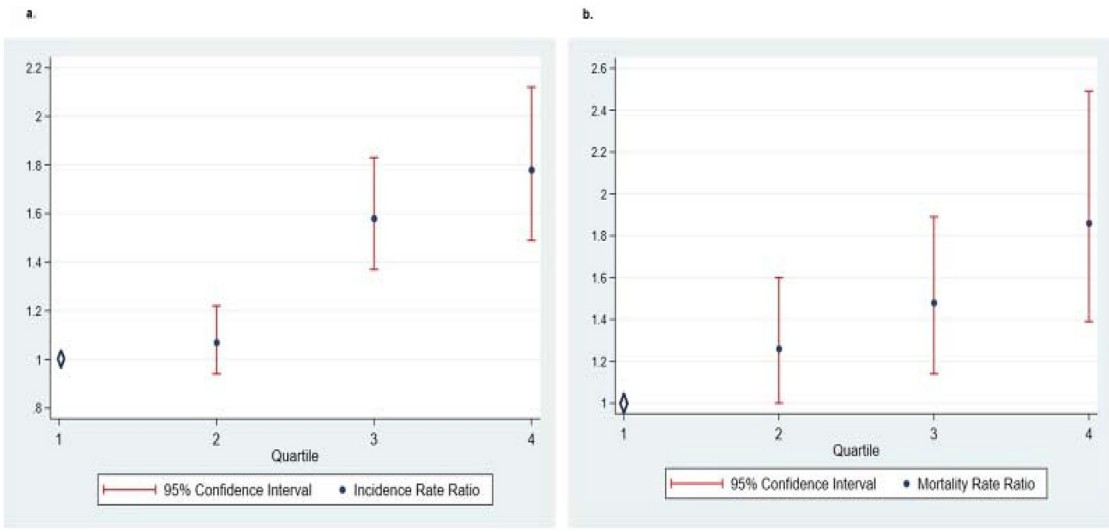

**Fig 1. Relative risk increase in incidence and mortality of Covid-19 with each quartile of percentage of households with poor housing conditions.**

**Table 3. Relative risk of county Covid-19 mortality as of April 21, 2020 related to percent households with poor housing conditions.**

| Model | Mortality Rate Ratio, MRR (95% Confidence Interval) |
|---|---|
| I- Percentage Households Living with Poor housing conditions | 1.63 (1.48–1.79) |
| II- Model I + Population Density and Test Density | 1.62 (1.47–1.78) |
| III- Model II + Demographics (Male, Age, White%) | 1.25 (1.12–1.40) |
| IV- Model III + Socioeconomic Factors (Median Household Income, % Residents with Lack of High School Education) | 1.41 (1.26–1.58) |
| V- Model IV + Respiratory Exposure Variables (Annual PM25 and Percentage of Residents who reported Smoking) | 1.49 (1.32–1.67) |
| VI- Model V + County Prevalence of Comorbidities (Diabetes and Obesity) | 1.44 (1.27–1.63) |
| VII- Model VI + Hospitalization Rate per 1,000 Medicare Beneficiaries (HF, Dysrhythmia, HTN, Ischemic Stroke, MI) | 1.44 (1.27–1.64) |
| VIII- Model VII + %Adults without insurance + Number of Hospitals with ER/county (Fully Adjusted Model) | 1.42 (1.25–1.61) |
| Sensitivity Analyses | |
| IX- Model VIII with Minority Demographics (Male, Age, Black%, Hispanic %, Asian%. Native Americans% in lieu of White%) | 1.37 (1.21–1.56) |
| X- Model VIII with poor housing conditions Quartiles as an Ordinal Variable | 1.22 (1.11–1.34) |
| XI- Model VIII for March 31st | 1.36 (1.10–1.67) |
| XII- Model VIII for April 10th | 1.31 (1.14–1.51) |

Note: Mortality Rate Ratios are from: Multilevel Generalized Linear Negative Binomial Models allowing random intercepts for States. A 5% increase in percent households with poor housing conditions represents approximately 1-SD change.

household income, lack of education and lack of health insurance, as the relative risk was only mildly changed and remained highly significant after adjusting for these factors.

Our findings have health policy implications as they identify a particularly vulnerable population to be at heightened risk and also the potential pathways for public health interventions during the current COVID-19 pandemic. In addition, our study adds to a robust body of evidence for other disease processes, which has shown that inadequate housing is a public health hazard especially in relation to infectious diseases and highlights the importance of finding short (e.g. better access to clean water and bathrooms) and long-term (e.g. overcrowding, cost) solutions to problems surrounding poor housing to help contain or mitigate the spread of COVID-19.

The strength of this study is that this is a nationwide report of 3135 counties across US, which allows for a large sample size and generalizability of our findings. Furthermore, to our knowledge, this is the first study to establish an association between the incidence and mortality of COVID-19 and poor housing conditions. The limitations of the study also merit consideration. First, the county-level covariate data utilized were from earlier time period, and hence may have weakened the strength of the associations. However, we utilized the most updated results publicly available. Furthermore, the assumption that county age structure and ethnic composition does not quickly change over the span of several years is the current approach shared by the US Census methodology (every 10 years). However, the consistent results after accounting for extensive list of covariates and various sensitivity analyses, support the robustness of the findings. Due to limitations of the data, we could not separate the distinct elements (e.g. overcrowding, cost, plumbing, kitchen) that comprised poor housing for better understanding of the problem and targeting of policies. This is also a cross-sectional ecological analysis and does not lend itself to causal inference. Finally, despite careful adjustments and inclusion of covariates, residual confounding cannot be excluded.

## Conclusion

In a nationwide analysis of US county-level data, counties with a higher percentage of households with poor housing had higher incidence of, and mortality associated with, COVID-19. These findings suggest targeted health policies to support individuals living in poor housing conditions should be considered in further efforts to mitigate adverse outcomes associated with COVID-19.

## Supporting information

**S1 File.**
(CSV)

## Acknowledgments

The views expressed in this paper represent the authors and not the Department of the Veterans Affairs.

## Author Contributions

**Conceptualization:** Khansa Ahmad, Sebhat Erqou, Wen-Chih Wu.

**Data curation:** Khansa Ahmad, Umair Nazir.

**Formal analysis:** Khansa Ahmad, Sebhat Erqou.

**Methodology:** Khansa Ahmad, Sebhat Erqou, Nishant Shah, Wen-Chih Wu.

**Supervision:** Sebhat Erqou, Wen-Chih Wu.

**Validation:** Khansa Ahmad.

**Visualization:** Khansa Ahmad.

**Writing – original draft:** Khansa Ahmad.

**Writing – review & editing:** Sebhat Erqou, Nishant Shah, Umair Nazir, Alan R. Morrison, Gaurav Choudhary, Wen-Chih Wu.

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
