## [Decision Letter · Decision Letter 0]

13 Jul 2020

PONE-D-20-15293

Association of Poor Housing Conditions with COVID-19 Incidence and Mortality Across US Counties

PLOS ONE

Dear Dr. Wu,

Thank you for submitting your manuscript to PLOS ONE. After careful consideration, we feel that it has merit but does not fully meet PLOS ONE’s publication criteria as it currently stands. Therefore, we invite you to submit a revised version of the manuscript that addresses the points raised during the review process.

I apologize for the delay; it has been very difficult to obtain reviewers during the pandemic given the abundance of manuscripts and manuscript review requests.  Please respond to the reviewer comments on a point-by-point basis and revise the manuscript accordingly.

We look forward to receiving your revised manuscript.

Kind regards,

Jeffrey Shaman

Academic Editor

PLOS ONE

Journal Requirements:

Reviewers' comments:

Reviewer's Responses to Questions

**Comments to the Author**

1. Is the manuscript technically sound, and do the data support the conclusions?

Reviewer #1: Partly

2. Has the statistical analysis been performed appropriately and rigorously? 

Reviewer #1: No

3. Have the authors made all data underlying the findings in their manuscript fully available?

Reviewer #1: Yes

4. Is the manuscript presented in an intelligible fashion and written in standard English?

Reviewer #1: Yes

5. Review Comments to the Author

Reviewer #1: Overall review and general recommendation:

Understanding within country disparities of COVID-19 cases and mortality is important for informing tailored approaches to the pandemic. The authors present a cross-sectional, ecological study of the association between poor housing conditions and outbreak intensity (number of cases and number of deaths) in 3135 US counties.

The paper addresses the important topic of structural determinants of COVID-19 and has the potential to be of high impact. However, there are several critical issues with the methods and interpretation. Therefore, I recommend major revisions. Below I detail my concerns.

Major comments:

1. It is unclear why number of cases and deaths are used instead of standardized rates. Presenting and comparing standardized measures would strengthen the paper.

2. Because the outcome does not seem to be standardized (is defined as number of deaths and infections), the interpretation of the outcome (IRR and MRR) is unclear. For example, a 5% increase in the percent of households with poor housing conditions is associated with a 50% higher risk of COVID-19 incidence. And yet, incidence seems to be defined differently in this paper – if I understand correctly, incidence is defined as number of deaths (and not deaths per population). Thus, the result could be interpreted as a 5% increase in the percent of households with poor housing conditions is associated with a 50% increase in the probability of each additional death per county (correct?).

On page 7, line 120, the authors state total population is used as the denominator, but the outcome seems to presented as counts (number of cases and number dead) throughout the paper. I recommend using rates or further expanding on the decision to use count data.

3. In adjusted analyses, the authors control for population density, but this won’t adjust (or standardize for comparison) for the underlying total population per county from which cases arise (due to different areas of the counties).

Minor comments:

1. Introduction: Overall, the Introduction is very brief. The rationale for the measurement of poor housing conditions is described in the Discussion. I think it would be helpful to readers to have this background at the beginning of the manuscript. In particular, the reason high housing costs is included would be informative, and seems distinct from the structural issues, such as shared toilet or kitchen facilities.

2. Page 8, lines 129-132: How were variables chosen for inclusion in the final model?

3. Page 15, lines 184-185: The inclusion of additional dates is a strength of this study. However, I would expect increasingly higher R (and thus exponentially more cases) in the counties with poorer housing conditions, but the finding is that the results are consistent across time. This could be further explored.

4. The Conclusion calls for targeted health policies to support individuals living in poor housing conditions. I recommend restructuring the Discussion a bit – moving the motivation for the exposure variables to the Introduction and/or Methods, and discussing the interpretation and implications of the findings in the Discussion. In particular, what kind of interventions might be successful? Are there interventions with demonstrated success in other countries – for example, meal delivery?

6. PLOS authors have the option to publish the peer review history of their article (what does this mean?). If published, this will include your full peer review and any attached files.

Reviewer #1: **Yes: **Danielle N. Poole

---

## [Author Response · Author response to Decision Letter 0]

21 Sep 2020

Dear Editor-in-Chief, PLoS One:

We thank the reviewer and the editors for their insightful comments and suggestions. We have tried to thoroughly address each of the questions/comments by the reviewer and we believe the manuscript has been made stronger by incorporating these changes. Please find enclosed a point-by-point response to the reviewer’s comments. 

Reviewer #1: Overall review and general recommendation:

Understanding within country disparities of COVID-19 cases and mortality is important for informing tailored approaches to the pandemic. The authors present a cross-sectional, ecological study of the association between poor housing conditions and outbreak intensity (number of cases and number of deaths) in 3135 US counties.

The paper addresses the important topic of structural determinants of COVID-19 and has the potential to be of high impact. However, there are several critical issues with the methods and interpretation. Therefore, I recommend major revisions. Below I detail my concerns.

Major comments:

1. It is unclear why number of cases and deaths are used instead of standardized rates. Presenting and comparing standardized measures would strengthen the paper.

Response - We apologize for our lack of clarity. Our analysis does represent standardized rates, standardized to the denominator population in each county. The outcome was a relative risk increase in incidence and mortality attributable to poor housing conditions. 

We utilized STATA® commands to get a standardized measure, by using the total population of the county as a denominator (page 7). Therefore, while the input variable is indeed the total number of cases or deaths in the county, the output is standardized by incorporating the total population in the equation. The “exposure(n)” in the command used (below) allows for calculation of rates within each county in the regression model.

We utilized the following STATA command

Equation 1- “meglm death poorhousing, exposure (n) || state:, covariance (unstructured) family (nbinomial) link (log) irr”

Where “death” is the number of deaths on the specified date in the county, “poorhousing” is the percentage of population living in poor housing conditions, “n” is the total population of the county. This allows to model the incidence of Covid-19 cases or deaths per given number of county population (e.g., per 100,000). The model output is a relative risk - either incidence rate ratio or mortality rate ratio.

2. Because the outcome does not seem to be standardized (is defined as number of deaths and infections), the interpretation of the outcome (IRR and MRR) is unclear. For example, a 5% increase in the percent of households with poor housing conditions is associated with a 50% higher risk of COVID-19 incidence. And yet, incidence seems to be defined differently in this paper – if I understand correctly, incidence is defined as number of deaths (and not deaths per population). Thus, the result could be interpreted as a 5% increase in the percent of households with poor housing conditions is associated with a 50% increase in the probability of each additional death per county (correct?).

On page 7, line 120, the authors state total population is used as the denominator, but the outcome seems to presented as counts (number of cases and number dead) throughout the paper. I recommend using rates or further expanding on the decision to use count data.

Response - We again apologize for the lack of clarity on the issue of standardized rate (standardized to the underlying population). As we indicated in the response to comment #1, the total population was incorporated into an equation as a denominator in modeling the incidence of case and deaths, thus the IRRs and MRRs reflect the relative risk increase in the incidence and mortality rates for 5% increase in the percent of households with poor housing conditions. Please see “Equation-1” in response to comment #1 above. We have now added appropriate units in lines 171, 176, 180 and 186 for further clarification. 

We have also added the following description to the methods section for further clarity :

“The outcomes of interest were incidence relative risk and mortality relative risk of COVID-19 attributable to poor housing condition. For this purpose, we obtained the COVID-19 cases and death data from the John Hopkins Coronavirus Resource Center, where it has been published for “public health, educational, and academic research purposes.”(2) We obtained the cumulative data for three dates of 10-day intervals: March 31st , April 10th and April 21st, which allowed us to test the robustness of our findings across three temporal cross sections of the reported data. April 21st data were utilized in the main analysis. These numbers of cases and deaths per county were subsequently incorporated into an equation with total population of a of each county as a denominator for standardization to incidence (cases/100,000) and mortality (deaths/100,000) in the regression modelling(26). Therefore, we were able to calculate the incident rate ratio and mortality rate ratio attributable to poor housing conditions. “

And therefore, the interpretation would be “each 5% increase in the percentage of households with poor housing conditions is associated with a 50% increase in incidence rate per given population (e.g. per 100,000)

3. In adjusted analyses, the authors control for population density, but this won’t adjust (or standardize for comparison) for the underlying total population per county from which cases arise (due to different areas of the counties).

Response – indeed as the reviewer points out rightly, adjusting for population density will not account for the total population per county. In addition to standardizing the regression equation output to the county population, we have also included population density as an additional adjustment in the regression equation to account for “crowding” at county level since our ongoing hypothesis is that transmission could be higher in highly densely populated areas.

Minor comments:

1. Introduction: Overall, the Introduction is very brief. The rationale for the measurement of poor housing conditions is described in the Discussion. I think it would be helpful to readers to have this background at the beginning of the manuscript. In particular, the reason high housing costs is included would be informative, and seems distinct from the structural issues, such as shared toilet or kitchen facilities.

Response- We thank the reviewer for this suggestion and have moved the rationale of individual components of poor housing conditions to the introduction: how high housing cost (more than 50% of the household income going towards housing cost) leads to a trade off from other aspects of living including health insurance and education and has been shown to contribute to delays in seeking healthcare. These four problems have been grouped together and reported as poor housing conditions by CDC. (In introduction: p.4, paragraph 2)

2. Page 8, lines 129-132: How were variables chosen for inclusion in the final model?

Response - All variables utilized in the study were informed by literature review and included in the final model. We have incorporated county level variables related to COVID-19 spread and outcomes based on literature referenced for each variable (Covariates in Materials and Methods).

3. Page 15, lines 184-185: The inclusion of additional dates is a strength of this study. However, I would expect increasingly higher R (and thus exponentially more cases) in the counties with poorer housing conditions, but the finding is that the results are consistent across time. This could be further explored.

Response – We agree that the incidence would be expected to increase with time (until effective measures are instituted and implemented) as is seen with number of cases and deaths in the descriptive statistics (Table 1). However, the relative risk of COVID-19 incidence and mortality related to poor housing conditions should remain fairly stable within the three 10-day intervals used in this study since there have not been significant effective strategies that disproportionately affected one group versus another of the main exposure in the COVID-19 risks during the study period. This is evidenced by the consistent IRR and MRR across different time points. 

4. The Conclusion calls for targeted health policies to support individuals living in poor housing conditions. I recommend restructuring the Discussion a bit – moving the motivation for the exposure variables to the Introduction and/or Methods, and discussing the interpretation and implications of the findings in the Discussion. In particular, what kind of interventions might be successful? Are there interventions with demonstrated success in other countries – for example, meal delivery?

Response- We have moved the motivation for the exposure variable (and each component thereof) to introduction as recommended. 

- We have added possible interventions to discussion, including increased education, improved ventilation and disinfection, increased availability of supplies and mobile bathrooms (lines 227-231, 247-248)

We hope you will find our revisions satisfactory and our manuscript will be acceptable for publication in your esteemed journal.

Respectfully,

Khansa Ahmad, MD & Wen-Chih Wu, MD

---

## [Editor Report · Decision Letter 1]

14 Oct 2020

Association of Poor Housing Conditions with COVID-19 Incidence and Mortality Across US Counties

PONE-D-20-15293R1

Dear Dr. Wu,

We’re pleased to inform you that your manuscript has been judged scientifically suitable for publication and will be formally accepted for publication once it meets all outstanding technical requirements.

Kind regards,

Jeffrey Shaman

Academic Editor

PLOS ONE
---

## [Editor Report · Acceptance letter]

19 Oct 2020

PONE-D-20-15293R1 

Association of Poor Housing Conditions with COVID-19 Incidence and Mortality Across US Counties. 

Dear Dr. Wu:

I'm pleased to inform you that your manuscript has been deemed suitable for publication in PLOS ONE. Congratulations! Your manuscript is now with our production department. 

Kind regards, 

on behalf of

Prof. Jeffrey Shaman 

Academic Editor

PLOS ONE